# Curating Violence: Reflecting on Race and Religion in Campaigns for Decolonizing the University in South Africa

Federico Settler

School of Religion, Philosophy and Classics, University of KwaZulu-Natal, Pietermaritzburg 3201, South Africa; settler@ukzn.ac.za

**Abstract:** During 2015 and 2016, staff and students at university campuses across South Africa embarked on two campaigns for decolonizing higher education, but the efforts were met with various forms of violent repression and rationalization of violence by state and private security services. In the face of the securatization of university campuses countrywide, ordinary mediums of teaching and learning proved inadequate for helping students reflect on their social reality, and similarly, public gatherings for socio-political deliberation and commentary became irregular because of the policing and surveillance of student protest action. By reflecting on the curation of three memorials and performances about seemingly racialized violence in this context, this article interrogates the meaning and the relation to the aesthetic, as well as the commentary on the context within which it is produced. Drawing on the work of Mbembe, Fanon, and Spivak as theoretical interlocutors with respect to how I understand violence, this article reflects on how three interdisciplinary curatorial events raise pedagogical challenges and opportunities for critical reflection in a context of repression. It was precisely through this interdisciplinary effort that the black body, violence, and context aligned to produce a public pedagogy on physical and representational violence. The three curatorial moments allowed for meaningful reflection on violence, resistance, religion, and the racialized self that not only drew attention to the artifacts and the performances but deliberately opened possibilities for a kind of public classroom where the discussion, articulation, and critique of violence is possible and productive.

**Keywords:** race; religion; violence; South Africa; decoloniality

## 1. Introduction

On 9 April 2015, the statue of Cecil John Rhodes was removed from the public area where it stood in the grounds of the University of Cape Town. This material and symbolic act was the culmination of a period of public deliberation and protest action by students on the campus. This moment also marked the beginning of a wider campaign of deliberation about the role of the public university in the postcolony and, in particular, the ways that black students and faculty find these institutions alienating. In fact, that language used by a student leader, Chumani Maxwele—who in an act of public protest threw a bucket of human faeces over the figure of Cecil John Rhodes a month previously—spoke to violence directed towards black students as they sought to make the university their home. Maxwele stated that " . . . there is a certain kind of violence that is done to a black person when everything around you reinforces the idea that this is not your world, and you are not welcome." This kind of symbolic violence (Bourdieu [1980] 1990, 2001; Fanon 1967), produced a discontent among South African students from university campuses across the country and gave birth to the largest public interest campaign in the country's democratic era. While students at other campuses had engaged in

similar regional protests for several years already, the student campaign known as #Rhodesmustfall and #Feesmustfall, were, respectively, oriented to address the transformation of the institutional cultures of universities perceived to be white-normative and to facilitate access to higher education for black students.

The protest actions from students assumed various forms during the two periods of academic disruptions that took place between early 2015 and the end of 2016 and drew a wide range of responses from supporters and critics (Ndelu et al. 2017). This period was also marked by the unexpected mass mobilization by activists through the use of social media, as well as transnational solidarity action reminiscent of the occupy movement, with explicit links to #blacklivesmatter across the Atlantic. This campaign was overt about its privileging of issues related to the alienation of people of colour in higher education or their sense of alienation in these institutional contexts. These recent campaigns are best understood as contemporary articulations of a longer history of protest against black people's exclusion from higher education in South Africa (Higham 2012), and growing feelings of discontent among black students and academics that, despite overcoming apartheid university curricula and institutional cultures, remain largely colonial and white-normative (Badat 2010). Thus, these campaigns produced several other innovative student actions in addition to the regular politics of spectacle (Habib 2016) that, among other things, took the form of confrontations between armed police and chanting students. At the bigger metropolitan city universities of Cape Town and Witwatersrand in Johannesburg, students occupied administrative buildings through sit-ins and occupation. At the University of Cape Town, the vice-chancellor's office building was renamed Azania House, and, at the University of Witwatersrand, the occupied building was named after anti-apartheid martyr Solomon Mahlangu. These occupations lasted for months at a time and saw students use them as their campus campaign centres, where they would invite regular speakers, hold workshops, and use them as a space into which to invite the university administration for meetings. At the University of Stellenbosch, students in the theology faculty took portraits of colonial and apartheid-era deans and turned them upside down or facing the wall in acts of defiance. Elsewhere, colonial and apartheid-era art and portraits were removed, damaged, and/or burned. At several campuses, students erected makeshift shanty homes to symbolically highlight the socio-economic realities of many students (Evans 2016), while, elsewhere, women students staged nude protests against police brutality. In her account of violence in the student campaign, Hlengiwe Ndlovu (2017) explained that " . . . the moment that we stepped into the protest, because as much we were displaying the intolerance of violence, at the same time, it was a form of resistance to say the very same woman's body is capable of ceasing fire and no one from national or local government could stop it." The women achieved their objective for the intervention; police officers withdrew and ceased fire against the shield of naked black bodies.

On provincial university campuses, students incorporated indigenous religious rituals into their protest action through the burning of mphepho (an indigenous herb burnt as incense) as a protective measure against police violence and the incorporating of their ancestors into their campaigns for decolonizing the university. In this context, religion offered an accessible and tangible register for marking out the sacred, legitimating the student protest action and linking the current struggle to a primordial ancestral struggle for recognition. The invocation of the ancestor through the burning of mphehpo as an indigenous ritual not only centred and incorporated the ancestral into the student campaigns, it also produced local articulations of what constitutes decolonial protest action. What I mean by this is that while occupations, sit-ins, and protest marches echo transnational civil society practices, the invocation of the ancestral disrupts regular ideas about resistance. The incorporation of the ancestral is more than merely instrumental and symbolic, and, through linking their current student campaign to a longer sacred struggle for recognition, these students activate the primordial as a register for legitimacy and action. In doing this, students used religious ritual as a mechanism to mediate and introduce new, indigenous, and decolonial registers through which they could produce protest action.

These efforts to incorporate the indigenous or to more meaningfully reflect on the lived reality of the African students that attend university must be viewed in relation to projects concerned with decolonizing the university. The decolonial project cannot be simply about changing the curriculum; it must also be about changing the institutional and social cultures of universities. Black students and scholars, as well as women and indigenous persons, have often felt out of place in these institutional contexts and have embodied experiences of being treated as space-invaders or intruders (Ahmed 2012; Mirza 2015), which, in turn, produce various kinds of imposter syndrome. The push-back against this long history of exclusion has taken various forms of campaigning for black academic associations, scholarship schemes for black students to enter postgraduate research, and so forth. While there has been formidable scholarship in the field of decoloniality, exemplified by the work of Nelson Maldonado-Torres, Ramon Grosfuguel, Linda Tuhiwai-Smith, Sabelo Ndlovu-Gatsheni, and Maria Lugones, it is only in recent years that students and faculty in South Africa had begun to grapple with the implications of decoloniality in campaigns such as #Rhodesmustfall and #Feesmustfall. In a presentation titled "Decolonising knowledge and the Question of the Archive," Achille Mbembe (2015, p. 6) notes that "when we say access, we are also talking about the creation of those conditions that will allow the black staff and student to say [about] the university: This is my home. I am not an outsider here. I do not have to beg or to apologize to be here. I belong here."

In the South African student campaigns of 2015 and 2016, one of the most contested issues was that of violence—violence by security personnel and public order police in their efforts to contain student action, as well as violent acts by students as they reportedly damaged university property. In many places, public campaigns have been met with violence, and it has become a taken-for-granted aspect of public protest culture (Duncan 2016). In South Africa, where at least 30 service-delivery protests occur daily, the violence that accompanies such protests has become such an everyday occurrence that the media barely reports on it any longer. Thus, as students on university campuses outside the metropolitan areas produced local chapters of the #Rhodesmustfall and #Feesmustfall campaign, we noticed that it was evident that a different kind of violence was reserved for students on largely black university campuses (Engh and Settler 2016). As a university lecturer of religion and race, I, together with colleagues from various disciplines, was outraged at the violence directed at our students; we, like our peers on other campuses, chose to support our students' rights to plan, structure, and state their protests. Our efforts to support the students on our campus resulted in a newspaper article in a national paper in late September 2016 with the title "Profs of protest: University staff accused of 'fuelling' plans to destabilise campuses." Despite this and other criticisms of us as academics, we defended our support of the students and, in an open letter published in October 2016, stated that:

> "Although we find the article untrue and full of spurious claims, we embrace the designation 'Professors of Protest,' despite the slanderous intent. If it means that we offer a space for free and critical dialogue about academic and financial exclusion; if it means decolonising the curriculum; if it means fighting, teaching and writing for social justice; if it means we put our bodies between students and security services to defend the right of our students to register their dissatisfaction, alienation and marginalisation; and, finally, if it means we defend the public university as a space for critical dialogue and exchange, then we are, proudly, Professors of Protest." (Settler et al. 2016)

As researchers and lecturers, we saw our role as defending the university as a space for critical dialogue and providing spaces for students to explore and examine their lived reality. As the repression of dissent persisted, and almost all avenues for deliberation among students had been 'banned' from campus, myself, together with three colleagues from visual art, drama, and anthropology, decided to offer a form of public pedagogy through art. It emerged as an organic effort to help students reflect on their reality, and so we began a process of reflection teaching, learning, speaking, and curating violence during the 2016 #FMF campaign at UKZN's Pietermaritzburg campus. Through this collaboration, we curated three public art events that not only addressed issues of violence on campus but also invited critical reflection and deliberation among students. Since these installations and performances did not

register as protest action with security services, they provided a safe space for students to meet, reflect, and exchange ideas.

## 2. Race and Violence in South African Student Politics

It goes without saying that reflections on, theories about, and discourses on violence emerge most pertinently in contexts of discontent, struggle, and repression. The encounters with violence have shaped much of the #FMF students' struggle—which has been variously defined as violence of exclusion and violence of dismissal enacted on black African persons.

As students, activists, and academics have sought to articulate the moral, ideological, and theoretical foundations of this student campaign, they have invoked the writing of theorists such as Frantz Fanon and Steve Biko because they believed these writings best speak to their lived condition. This history of entanglements of Fanon and Biko in the South African struggle is best captured in Nigel Gibson's *Fanon Practices in South Africa*, in which he maps the various ways in which Fanon has been incorporated and animated in various South African struggles after apartheid. Gibson's attempt to engage "Fanon's untidy dialectic with new realities" (Gibson 2011, p. xvii) leads him to survey Fanon's influence over struggles for recognition from Steve Biko's black consciousness in 1970's South Africa to the struggle for basic service by shack-dwellers associations in post-apartheid South Africa. This survey includes Biko's 1970 essay *Black Souls in White Skins*, echoing Fanon's *Black Skin, White Masks*, culminates with an examination of the contemporary shack-dwellers association, *Abahlali baseMjondolo*, as an embodiment of Fanon's pitfalls of national culture in its siding with the poor and distancing itself from the postcolonial state (Gibson 2011). Reflecting on the same social movements, Kerry Chance (2018) eloquently argues in her *Living politics in South Africa's Urban Shacklands* that fire, water, land, and pollution frames the material realities of the urban poor in South Africa, and that fire has been deployed by the poor as a weapon of resistance against government practices. In their various protest actions against the lack of basic services, these movements appeared to rely on a Fanonian register of violence to legitimise their particular forms of protest action, including road blockages, performative demonstrations with traditional weapons, and arson. In translating the Fanonian practices of the urban poor to the university context, students sought to read Fanon's views of violence; on three occasions in 2016, I was invited to facilitate seminars on Fanon, Race, and Violence. In these contexts, students cited Fanon's assertion that:

> [d]ecolonization is the meeting of two forces, opposed to each other by their very nature
> . . . [t]heir first encounter was marked by violence and their existence together (Fanon 1968,
> p. 36)

*Wretched of the Earth* is run through with references to violence enacted on the bodies and communities of colonised people, with Fanon asserting that while violence in this context is the product of colonialism, and a collective catharsis is necessary for the colonised subject to violently rid themselves of colonial rule (Fanon 1968, p. 42). While many student activists found resonance in Fanon's notion that "violence . . . frees the native from his inferiority complex and from his despair and inaction; it makes him fearless and restores his self-respect" (Fanon 1968, p. 94), many were keen to draw attention to effects of sustained police and social violence on their sense of confidence and feeling part of the university community. During these periods, students often articulated their campaigns as struggle and combat. For example, at one university, a student reported: "You know the police, they simply come and start firing rubber bullets, teargases, and water cannons to disperse students. This is what makes things worse" (Protest student 15, Tswane University, November 2016); 12 months previously, the Student Council, frustrated by failed talks with the university administration in a letter to the executive of the same university, wrote "We are ready to die for a black child to be granted his or her right to education" (Vilakazi 2017, p. 54).

These sentiments of combat and the courting of violence were shared by those involved in the student campaigns across the country. Students sought to access various ideological and theoretical

registers for explaining and analysing their experiences of violence. This included the inconsistent and ambivalent courting of theories such as Mbembe's necropolitics because it points to the suggestion that "technologies of destruction have become more tactile, more anatomical and sensorial, in a context in which the choice is between life and death" (Mbembe 2003, p. 34), as well as Spivak's epistemic violence, insofar as it drew attention to institutional practices and structures of knowledge that silences the experiences of the (formerly) colonised. While most students in these campaigns had little working knowledge of Spivak or Bourdieu's symbolic violence, they believed that institutions of higher learning and accompanying knowledge regimes were produced to control and exclude the Other. They further believed that destabilizing or confronting such endemic violence was only possible through riots and mutinies (Spivak 1999). The recent student campaigns in South Africa such as #RhodesMustFall and #FMF represent such moments of contestation, critique, and theorizing about violence—partly because they are also a convergence of material violence, state violence, systemic violence, institutional violence, and epistemic violence.

In the aftermath, a range of texts have been produced on the social, political, and economic impact of the #Rhodesmustfall and #Feesmustfall campaigns. This growing field of inquiry proliferated across southern Africa, producing texts such as *Fees Must Fall: Student Revolt, Decolonisation and Governance in South Africa* edited by Susan Booysen (2016), *Rebels and Rage* by Adam Habib (2019), *As by Fire: The End of the South African University* by Jonathan Jansen (2017), and *#FeesMustFall and Youth Mobilisation in South Africa: Reform or Revolution?* by Musawenkosi W Ndlovu (2017). Overall, these texts are the result of work by political scientists and educationists who variously try to account for either the forces that resulted in these campaigns or to frame and propose possible futures for the African University. One volume that resonates with our curatorial efforts is the student publication *Rioting & Writing: Diaries of the Wits Fallists* (2017), wherein students who were active in the movement produced personal narratives incorporating body, gender, protest tactics and solidarity with groups beyond the university. This latter volume attempts to both give an account from the perspective of the students as well as serve as a popular pedagogical effort whereby students were afforded the chance to think through the issues they wanted to write about and publish, as well as the broader purpose(s) of discussing and disseminating their ideas. On our campus, we constructed three curatorial moments to similarly help students and staff think through the meaning of violence in the context of the university.

Framing these student campaigns as violent, seditious, and disruptive was not simply a product of media imaginaries, although it played its part. What is beyond question is the fact that violence happened. It was enacted upon our students. Students committed arson through the burning buildings, books, and cars. Students were arrested, shot, and brutalized, and staff was threatened. While much of these acts of violence, arson, and vandalism are contested, that it happened is not disputed (Grassow and Le Bruyns 2017). Until late 2016, many South African university campuses remained occupied by police and security services, with many universities having established a police command centre on campus, supported by a court interdict. As we witnessed a change of rhetoric from speaking of students as clients to speaking of students as criminals, I wrote a piece with a colleague commenting on the violence perpetrated on predominantly black university campuses. We concluded that:

> "Witnessing the enthusiasm and "glee" with which police and security staff enforce the interdict, it is hard not to agree with students' assertions that this is a level of intimidation and brutality by police and dismissal by university management is reserved for black students at provincial campuses."

The escalation of violent repression by the state, university managers, and security services was met, nationally, with an escalation of resistance (Booysen 2016). As students' arrests increased and raids on student residences became commonplace, many sought refuge in local churches and in other places, religious leaders and staff presented themselves as protective shields as they placed their bodies between protesting students and police (Munusamy 2016). For much of the period of the campaign, especially during late 2015 and early 2016, our social media was saturated with images of violence as we adjusted to increased securitization or militarization of campuses. Students, staff, and the

general public were glued to their smartphones and laptops, producing, reporting and consuming "protest porn."

A widely debated aspect of violence in the student campaigns was the issue of arson, with most newspapers reporting largely on what students burnt and very little about police brutality. Kerry Chance, in her book, *Living Politics in South Africa's Urban Shacklands,* reminds us that arson has been commonplace in South Africa struggle for resistance and social self-determination. This opens the possibility of imagining violence or arson in protest cultures as more than just deviant or undesirable. Chance (2018) argues that in the face of armed authority, fire is an accessible form of resistance—it is ignition in the hands of the poor, where "between life and death, fire draws its ultimate line of difference."

Attempts by students and staff to speak and think through the violence, arson, and intimidation were largely met with silence, anxiety, and a reluctance to speak publicly, and there appeared to be a lack of a suitable language to articulate the complex engagements with violence that we as a community—staff and students—experienced. On most campuses, the various court-mandated interdicts prevented students from gathering in public spaces across the campuses in groups of more than five. Students were prevented by management from meeting and speaking about what was happening. Moreover, we/they also lacked the language to speak or make manifest our collective/individual trauma—whether in a desire to speak to Mbembe's notion of the necropolitical threats to life or to Spivak's notion of epistemological violence of exclusion and representations of students as criminal. We were numbed and muted. We could not feel, and we could not speak.

## 2.1. Three Curatorial Moments

During late 2016, at the height of the repression by the state, courts, and university administrators, conditions on my local university campus, as was true also for most university campuses across the country, meant that university programmes continued under highly securitised conditions. Students were barred from meeting in groups outside lecture and seminar venues, and all main teaching venues were guarded by security staff who controlled access. Those who contravened these conditions and engaged in group protests, occupations, or sit-ins were arrested and imprisoned. For two weeks during the third academic term of 2016, a group of concerned faculty invited students—any students—to attend seminars that we convened at a large teaching theatre. We used these seminars as opportunities for students to meet and talk about their concerns in a safe space. These learning events included poster design and printing, writing blog posts, contextual bible studies, and site-specific theatre performances. Soon our pedagogical efforts were curtailed, and concerned staff were denied the use of this teaching theatre, thus leaving no space for students to meet or discuss their needs, concerns, or collective strategies related to the two national student campaigns.

At our campus, the academic programme was either suspended as a strategic effort to quell disruptive protests (Grassow and Le Bruyns 2017), or the academic programme we claimed to continue as normal, with extensive surveillance by police and security staff controlling access to campus and key buildings. In the face of this severe repression, students moved about the university anxious, angry and traumatised. In an effort to craft as collaboration, myself and three colleagues from visual art, drama, and anthropology, as well as some student leaders, began a process that saw us curating three events that gave critical accounts of the violence that the students experienced, provided commentary on the adverse impact of the violence on the university community, and provided a space for engagement and deliberation in the midst of a violently repressive context. The decisions related to the type of curated events was the result of organic but careful planning, insofar as the three events were not planned together but emerged in relation to each other. As academic staff, our first effort was to create a space for public reflection about the 'new normal,' and while the subsequent performance and installation were intended as a critical commentary, they also served as spaces for student to voluntarily participate in the curated event through registering their experiences or thoughts on the violence that characterized the student campaigns. While we enjoyed a shared idea of curating a selecting,

organizing, and representing ideas, images, and events, in this context we approached curating as a contextual practice concerned with framing the political economy of art (Greenberg et al. 1996, 2ff; Lacy 1995) or what Chantal Mouffe (2013) regards as the radical potential of the artistic for shaping hegemonic struggles for public space. In this regard these curatorial events had the dual functions of being a commentary and critique on the violence inflicted on students, as well as a catalyst for a decolonial public pedagogy or border pedagogy. They contributed to the ongoing deliberation about the role and significance violence in the student campaigns.

*2.2. Prisoners' Memorial*

Between 22 September and 10 October 2016, 21 students from the Pietermaritzburg campus of UKZN were arrested and imprisoned, charged with public disorder and contravening the university's interdicted against meeting in group in public. The university announced that the trouble-makers had been removed and that the academic programme would return to or continue as normal (Grassow and Le Bruyns 2017). Nothing about the state of security and imprisonment of those students was normal. Despite the fact that staff and students had been prevented from meeting and speaking the arrests as a community, I wanted to make visible the trauma and absence of students in prison. With colleagues, I built a memorial—it was constructed at home and installed on the main campus, the site of the then-banned meetings, exclusions, and arrests (Figure 1). The memorial mimicked a cemetery and took the form of 21 flattened black refuse bags secured to the ground in neat rows on the grass lawn in front of the main campus library. On each flattened plastic bag, there was a sheet of paper detailing the name of an arrested student, where they were imprisoned, and the number of days they had been incarcerated.

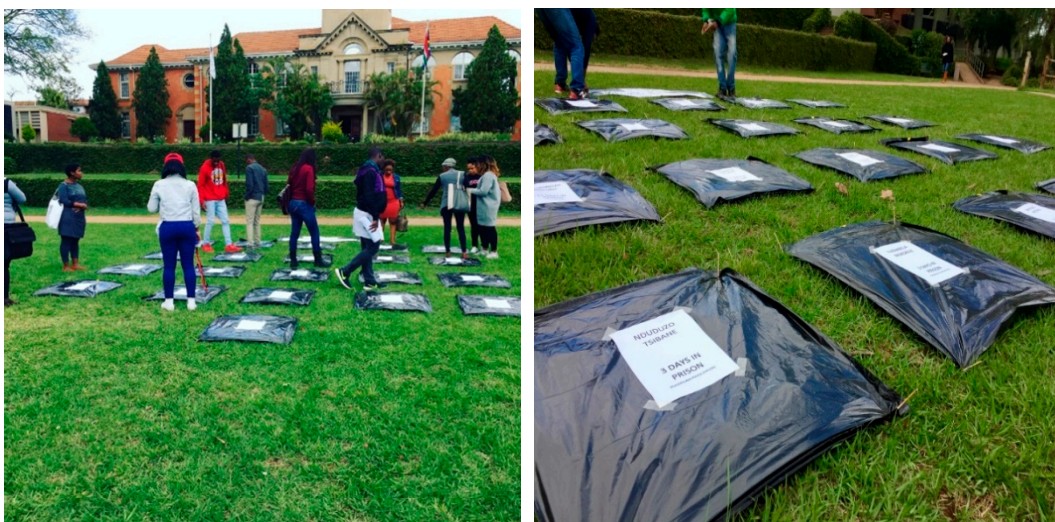

**Figure 1.** Prisoners' Memorial.

This memorial had an immediate impact, and it drew students' curiosity. As an artistic installation, it was not regulated under the court interdict. Students from across the campus came and read or stood in silence to remember their colleagues. Several students sat down and cried, while others expressed shock that fellow students had then been imprisoned for more than 10 days at a regional high-security prison. As faculty, we did not intervene nor seek to direct student reflections or comments. Students lingered, occupied the space, and left; others would return with friends. The popular sacrality of the memorial space kept security staff at a distance. Though it was a sombre space, detailing student biographies and number of days incarcerated, it was also a space for critical reflection and deliberation between students and faculty invested in the outcome of the campaign. For those students who viewed the #FMF campaign from a careful distance, worried about the violence of the police and alienated by the violence of some students, this memorial brought into focus political and material dimension of

student's imprisonment and the institutionalised violence. We observed students and faculty register outrage at the fact that their peers were arrested and imprisoned for speaking up about the conditions of higher education. A few noted that the memorial itself was subversive insofar as it provided a space for students to gather and reflect on issues of exclusion and alienation in higher education.

Curating this memorial was a deliberate effort to speak to the social death that comes from the imprisonment of dissidents and protestors. Where ordinarily war or conflict sites and monuments memorialise entanglements with violence through producing public sites as "vehicles of memory," which tend to create martyrs and victims (Jelin 2007; Duncan 2009), this curatorial effort drew attention to the threat of death. Further, it not simply memorialised the absent but also humanised those imprisoned, creating lines of affinity and affect between those absent and those standing in the prisoners' memorial. Through creating a bounded space, with imagined sacrality, we facilitated a space for learning and reflection where students brought their respective experiences of violence on campus into conversation with the real and lived realities of both the prisoners, as well as other students amidst the memorial—thus creating a participatory learning space.

### 2.3. Poetics of Protest

Several weeks after curating the prisoner's memorial, we hosted Koleka Putuma, a nationally renowned poet. Together with a group of student leaders, we agreed to use one of our regular monthly mentoring meetings for black staff, students, artists, and activists as a space for students to meet. We conjured up *Poetics of Protest*—an event where Koleka Putuma performed her poetry and invited students to read their own writing about the conditions on campus during the preceding months. The event (Figure 2) ended with her doing a final series of poems and group exercise on black worlds and interiors. This was conceived as a space for students to speak, and attempt to find words and to be offered space to speak into. The event incorporated poets and students from across the campus and a local Zulu (vernacular) poetry collective.

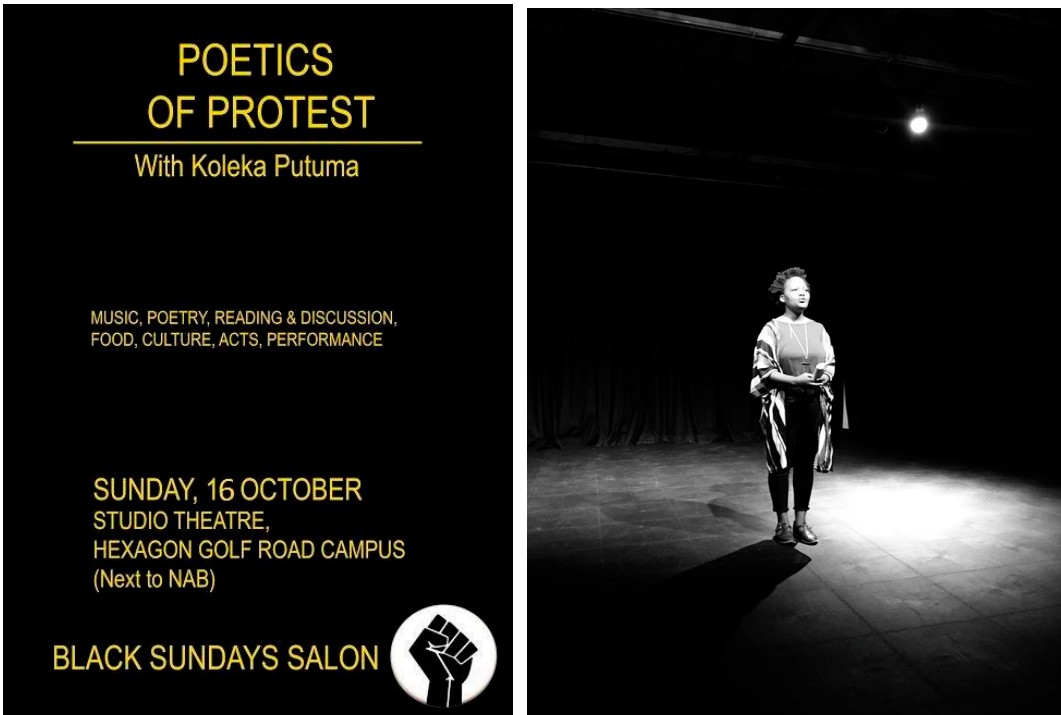

**Figure 2.** Poetics of Protest.

In a context where the politics of spectacle and performance had currency (Habib 2016), the invitation to participate in a poetry session was met with skepticism among most students. Nevertheless,

*Poetics of Protest* gained momentum as we sought to facilitate alternative modes of protest, speaking, and thinking—and it was well attended because it provided a space for students to meet in large numbers and speak publicly about decolonising the university and the repressive measure of the state and university leadership. It forced academic staff and students to engage in forms of resistance and subversion that did not involve, nor bait, direct violence from the police and security services. Putuma's poetry, now published in the volume *Collective Amnesia*, resonated with students' social reality drawing strong linkages between the symbolic and epistemic violence of being black on South African university campus in 2015 and the generally dismal condition of black people's lives 25 years after the advent of democracy. Putuma's poem "Water" resonated with many:

- Yet every time our skin goes under,
- it's as if the reeds remember that they were once chains,
- and the water, restless, wishes it could spew all the slaves and ships onto shore,
- whole as they boarded, sailed and sunk.
- Their tears are what have turned the ocean salty,
- this is why our irises burn every time we go under. (Putuma 2017, pp. 96–100)

This, an excerpt, from the poem, was greeted with raucous support and laughter captured the mood of alienation and frustration among the students, and when another of Putuma's poems offered a critique of the violence against women in the #Feesmustfall and #rhodesmustfall movements with her poem "On Black Solidarity," it was greeted with a sombre discomfort. The poem opens as follows:

- Black solidarity does not include making my spine a doormat
- so that you can stand or have a backbone
- Black solidarity as the expense of a black womxn's anything, is a farce, a rip-off.
- The kind of violence they shred into laughter at the police station
- And replay in front of you just to make sure you got the joke (Putuma 2017, pp. 80–82)

After what seemed like weeks of running battles, police raids, and prison visits, students were battle-fatigued, and this poetry session afforded students a sanctuary from violence and surveillance, as well as a space for reflection. While, on the one hand, the poetry session allowed space for recognition that opened-up some wounds of pain and anger, the cathartic poetics of protest opened another avenue for deliberation about the meaning and experiences of violence. Where the *Poetics of Protest* was a departure from the Prisoner's Memorial was in the fact that it not only drew attention to violence enacted on black students by the police and security service but also to the various ways that through design and execution of struggle, the student movement at times mimicked the forms of violence that they sought to condemn (Booysen 2016). We would later learn that following the event, students gathered in smaller groups in the residences and homes to continue the discussion about the relationship between decolonizing higher education and the struggle with the broader social struggle for human dignity in the communities they come from. "On Black Solidarity" opened up the possibility of a critique related to violence enacted by activists, and this became evident from the efforts by queer and gender activists not just critiquing institutional patriarchy but also patriarchal forms of struggle, and its entanglement with particular forms of violence (Ndlovu 2017).

## 2.4. Public Sculpture

The third curatorial moment, a public sculpture, forced us to think more deeply about the pedagogic intent of our efforts as resistance art, insofar as we hoped that it would present a response to the conditions of violence, a piece that would provoke students to reflect on and speak about the violence. We produced a public sculpture that was a commentary on the collective experience of violence and arson, to mourn losses—of innocence and things, to remember trauma, fire, and fear—and the abnormality of our existence made 'legal' by the interdict. Four staff members developed the sculpture (Figure 3), which was made up of steel-frames of desks, barbed wire, with slogan printed

t-shirts, a mirror, and burned textbooks on a bed of salt. The sculpture of charred texted books, clothes and twisted metal was meant to capture the twisted and entangled relationship of students to the university, and that it was covered in barbed-wire represented the restriction and repression of student protests. In a highly symbolic move, we placed a mirror in the centre of the piece as a way to incorporate the student-viewers into the sculpture. It provided a space of recognition, exchange, and reflection. No commentary accompanied the sculpture and we left it up to each student to leave their mark in the salt, or on the chalkboard. Many sat and discussed the meaning of the sculpture with their friends. The paradox of this curating of violence and fire, on a campus under repression was not lost on the students. Many of them were anxious that we or they would get into trouble, but we explained that we simply presented a commentary on what we observed about social and institutional relations on campuses across the country.

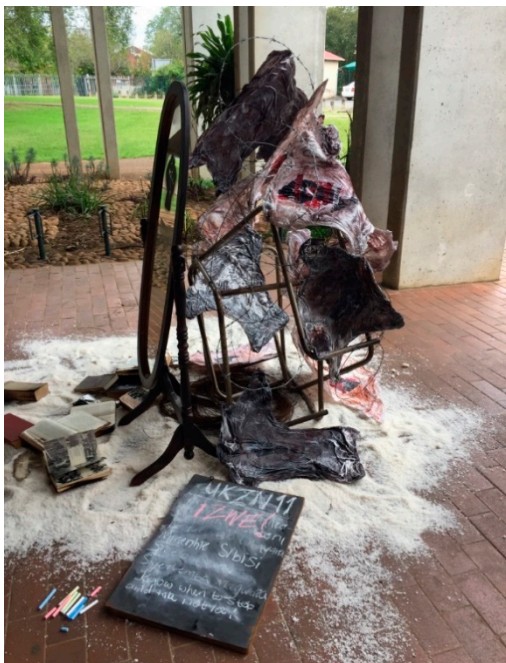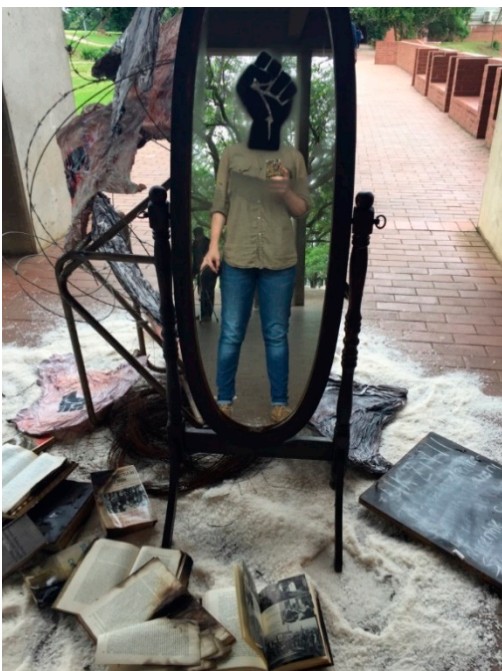

**Figure 3.** Public Sculpture.

Originally, we intended to burn the sculpture as a final act, but this was not permitted by the administration. As a commentary on violence and arson, it was firstly intended to disrupt ideas about where learning might take place through erecting the sculpture as a public classroom. Secondly, it sought to incorporate the context and students' experiences of violence into the learning space. At an epistemological level, it was meant to raise questions about what knowledge is valued and what life discourses are centred in the exchange between lecturers and students. The sculpture served simultaneously as a commentary on violence and a critique of violence, thus making possible nuanced and inclusive deliberation about violence that does not simply rely on binary articulation violence as diminishing (Arendt 1969) or liberative (Fanon 1968). It is my view that the public erection of the sculpture created sentiments of sacrality through placing students lived experience at the centre of a public pedagogical act that allowed learning to be continuous beyond what we as creators imagined.

*2.5. Border Pedagogy as Public Pedagogy*

In their examination of how to cultivate Walter Mignolo's border thinking among Mexican students, Cervantes-Soon and Carrillo (2016) argue that border pedagogy privileges the epistemologies and ways of being in the world of those who find themselves in colonial, social, and institutional borderlands. As they explore their students sense of alienation by and within mainstream educational institutions,

Cervantes-Soon and Carrillo argue that border thinking "instead constitutes a potentially radical way of re-imagining knowledge and educational practices for oppositional social transformation" (Cervantes-Soon and Carrillo 2016, p. 282). It is premised on the idea that students are possessors of knowledge, and that pedagogy's aim is to help student locate themselves within their own social context, with the view to activate new knowledge registers born from their lived experiences. In the context of the South African student campaigns of 2015 and 2016, I am not claiming that we decolonized the curriculum, but, in the context of violence and repression, we were by necessity required to work with an anti-imperialist model of teaching and learning. As such, these three curatorial moments not only disrupted the relationship between academic facilitator and students but opened new registers for thinking, and speaking about violence in the context of the university.

With regard to the experiences of fire, arson, and violence, black students have deep and intimate archives of knowledge that predate their coming to university. We learned that students possessed reservoirs of knowledge with respect to the power of fire and violence as a way to draw a line in the sand. For some, burning barricades in the street was a scream against poverty and alienation, while, for others, direct conflict was a mere extension of a longer history of intimate and institutional violence. Further, the manner in which many young black men and women move in social space betrays their acute awareness of being subject to particular regimes of surveillance—reinforcing their sense of alienation, of being out of place.

This reality resonates with Cervantes-Soon and Carrillo's argument that "By recognising these subaltern knowledges of border thinking, border pedagogy repositions people on the margins as creators, thinkers, and knowers. This constitutes the very condition of possibility as youth are given the opportunity to reclaim their agency and challenge dominant and Eurocentric intellectual thought in creative ways" (Cervantes-Soon and Carrillo 2016, p. 286). Similarly, Geoffrey Harpham, in his (Harpham 2005) *New Literary History*, reminds us that no subject in the humanities can satisfactorily address a social phenomenon, such as violence, as they are all concerned concern fundamental issues relating to humanity; we, in the humanities, must see as the natural sponsor of the debates and controversies that frame such issues. Together with my colleagues, I found that it was precisely through drawing on different disciplinary traditions that we were able to produce a range of publicly curated events that made possible varied ways of representing violence and enabled us to facilitate critical, gendered, and indigenous deliberations about violence.

As we attempted to nurture students subaltern knowledges, we drew on registers in art, drama, religion, politics, and education to facilitate the three curatorial moments also as a way to explore the relationship between knowledge and power in the context of violence against black students. Lucy Lippard (1997) suggests that the landscape and vernacular of the art object are best understood in terms of the space, or context within which we live. Walter Benjamin (2005), on the other hand, suggests that art renders accessible what was previously distant or *unreachable.* Like art, religion emerged as a resource in sanctioning the student's campaigns and legitimating violence, not an uncommon practice in South Africa (Chidester 1992). In particular, the sentiments of the ancestral invoked through the prisoner's memorial and the poetics of praise singing (imbongi) located the ancestral and the religious at the heart of the curatorial and pedagogic effort. Thus, religion emerged as not just legitimating violence but also as a register through which to understand, interpret, and make sense of violence in the context of the student campaigns for decolonizing South African universities. As such, the university became a landscape wherein which black students' told their stories of struggle and biographies as space invaders. What these curatorial efforts made possible were reflections on violence as ways to mediate subjectivities, to regulate access, and to register protest.

However, while these experiences of racialized violence as archives of knowledge are excluded from the regular knowledge economy of the university, the violence of protests and the desire to obliterate through fire must be examined as a way to reflect on what it represents. To dismiss violence and arson by students or protestors as socially deviant and undesirable closes the possibility to speak, teach, and theorise about important aspects of black students' experiences in ways that take it as

serious and valued. This foreclosure itself is a is a kind of epistemic violence itself. The decolonization of higher education requires that we centre those black experiences and experiences of women that move beyond the victim-villain trope—those experiences and knowledge registers that disrupt our comfort. Through curating violence, we hoped to place other ways of knowing, knowledge registers pertaining to violence at the centre of our pedagogical deliberations, as a way to privilege our students' experiences of violence, racism, and the ancestral over the supposedly universal white western ways of knowing and being, to produce a border pedagogy.

## 3. Conclusions: Curating Violence

The particular pedagogy pursued in the development of the three curatorial moments elaborated above was made possible by converging registers of meaning-making from religion, social science, and the arts. In an effort to help students reflect on their experience of violence and their efforts to decolonise the university—through deracializing and queering the curricula and making a democratic classroom—these curatorial moments were facilitated to interrogate the relations between power, dominance, and knowledge. Especially in a context where students were denied opportunities for meeting in large groups to reflect on and deliberate around their experiences of violent repression, these curatorial moments provided not just a space for meeting but also sought to help student think through their experience of violence, to reflect on the impact of violence, and to honestly discuss their ambitions with respect to deploying violence as an anti-colonial measure.

As academics and students, we were conscious that dealing with the topic of violence and arson on our campuses had become an unspeakably hot topic, so to speak, because it had emerged as a container for a wide range of anxieties, anger, and ambitions. On the one hand, it represented the animation of shackland protest strategies such as burning barricades on university campuses where it was alien, and, on the other hand, it scripted new discourses of protest cultures while closing down broad-based discussion of the meaning of violence. Through offering three differently curated events related to the issue of violence, the hope was to facilitate a public border pedagogy that would allow students to reflect on their experience and practices with the view to socially transform the university. This was made possible particularly because, on the one hand, art as social commentary and representation emerged as an acceptable medium through which to critique the repressive conditions, while religion, on the other hand, provided an accessible and indigenous register for making meaning. In the South African context, the invocation of the ancestral through the burning of incense or through funerary rites or depictions such as the prisoner's memorial entices popular piety. While the memorial invokes the necropolitical threat of death, it also humanised the absent and imprisoned students. In this sense, it was more than a memorial. Likewise, the praise-singing of fallen heroes and long lost slave ancestors echoed in the *Poetics of Protest* connected the contemporary student struggle within a longer primordial struggle for recognition. The recognition of these accessible, local religious rituals provided students with a register within which to make sense of an articulate their understanding of violence and about the transgressive and productive power of fire.

Further, I would argue that the public border pedagogy was made possible through the process of curating as a continuous deliberating process. Unlike the memorialisation of violence and its making of martyrs, curating violence emerged as an ongoing participatory pedagogical process that allowed staff and students to think through the intersection of knowledge and power with violence and resistance. The curated events invited more than just a gaze, insofar as all three events invited participation or presence. In order to read the prisoner biographies on the prisoner's memorial, student had to enter a seemingly funerary space, erected on the site of student clashes with police. Similarly, the *Poetics of Protest* and the Public Sculpture disrupted the seemingly normal order of campus through inviting students to enter into proximity with a curatorial event that overtly referenced violence against students.

Finally, all three curatorial moments engaged registers of the sacred and the artistic to make possible deliberations about violence otherwise not possible. This allowed students to reflect on and

speak about violence and to develop language, theories, and competencies related to their experience of violence that does not only rely on the thoughts and words of Fanon, Bourdieu, Spivak, or Mbembe but one that also draws on indigenous registers of sacrality. In this regard, the three curated events opened border pedagogical discourses about violence as not just material or instrumental but also about violence as sacred and as mediation over who and how to frame the decolonial project.

**Funding:** This research received no external funding.

**Acknowledgments:** The author extends recognition and appreciation to colleagues from University of KwaZulu-Natal for their collaboration in developing the final art installation in this series of three curatorial events. They are Tamantha Hammerschlag (Drama), Wayne Reddiar (Visual Art) and Mari Haugaa Engh (Anthropology).

**Conflicts of Interest:** This research received no external funding.

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
