# Peer review of "Curating Violence: Reflecting on Race and Religion in Campaigns for Decolonizing the University in South Africa"

_religions, doi:10.3390/rel10050310_

Round 1
Reviewer 1 Report
This is an excellent article, just the kind of work one would hope to see published in the field of the study of religions and the humanities more generally. There are only a few minor elements that in my view require either development or editorial work prior to publication.
With regard to the former, I think that certain ideas are left undertheorised or at least rendered more explicit than they currently are, e.g. 'decolonial protest action': what marks not only the student protest actions as decolonial (this much was evident) but also the curatorial interventions? Further, I would like to see a more concrete discussion of the interdisciplinary nature of the interventions, how this too may be read in a decolonial frame as resisting the silo-ing of disciplinary formations and the rigidity of institutional structures which mandate a certain kind of 'belonging' that is antithetical to that suggested by Mbembe. In many ways, one could read unyielding disciplinarity as itself a form of securitisation and policing of thought. Further on this point, it would have been useful to read more about how the curatorial team came together and how the process of developing the three different memorials came together. This latter point, however, is not necessary for the paper's publication but would simply add value and context. The abstract is a little misleading inasmuch as it states the paper 'draws on the work of Mbembe, Fanon, and Spivak...etc.'. Rather this work appears to have been a preliminary framework that by the end of the paper is reframed to suggest that student learning 'does not primarily rely on knowledge of Fanon etc.'. I think it would be useful to have a sentence or two making more explicit the pedagogical values and processes inscribed in this respect.
On the editorial front, the paper has quite a few typos, incomplete sentences, missing words, etc. that should be cleaned up before submission.
I enthusiastically recommend publication subject to these few issues being addressed.
Author Response
Response to Reviewer 1
Thank you for the attentive and detailed review (17 April 2019). Herewith my response:
1. certain ideas are left under-theorised or at least rendered more explicit than they currently are, e.g. 'decolonial protest action': what marks not only the student protest actions as decolonial (this much was evident) but also the curatorial interventions?
Response: in the introduction to the article, I have inserted two sets of remarks related to (1) the fact that the student campaigns form part of a long effort for re cognition and inclusion of people of colour into higher education in South Africa, and (2) that this campaign incorporated local and indigenous form of resistance. In this regard I set out to show how drawing on indigenous ritual and practices as a strategy for protest action, also disrupts ideas about what constitutes protest action. See lines 49-55, and lines 83-89
2. a more concrete discussion of the interdisciplinary nature of the interventions, how this too may be read in a decolonial frame as resisting the silo-ing of disciplinary formations.
Response: while the limits of disciplinary silos are addressed in my discussion of the humanities (Harpham 2005), I added notes on the explicit benefits of working within an interdisciplinary method, and its value for facilitating decolonial deliberations. See lines 460-463
3. how the curatorial team came together and how the process of developing the three different memorials came together.
Response: At the end of the introduction of the article, I provided a short account of the ways that academic staff more generally got involved in support of the student campaigns generally, as well as how we, as four academic came together to start of a process public pedagogy through art (see lines 134-144). The pedagogical approach in developing the three curatorial events has been elaborated in the article. (See lines 283-289)
4. The abstract is a little misleading inasmuch as it states the paper 'draws on the work of Mbembe, Fanon, and Spivak...etc.'. Rather this work appears to have been a preliminary framework that by the end of the paper is reframed to suggest that student learning 'does not primarily rely on knowledge of Fanon etc.'. I think it would be useful to have a sentence or two making more explicit the pedagogical values and processes inscribed in this respect.
Response: I have revised the abstract to adjust the claims made Mbembe, Fanon and Spivak as theoretical interlocutors that help with framing violence in the essay – and to more accurately reflect their place in the article (see lines 15-17). In the end they do become the less visible backdrop against which teaching and learning about violence, race and the sacred might take place – through curated events and students’ reception thereof. In the conclusion, I have also revised some notes about the pedagogical value and processes. (see 532-535)
5. On the editorial front, the paper has quite a few typos, incomplete sentences, missing words, etc. that should be cleaned up before submission.
Response: The article has been thoroughly copy-edited.
Reviewer 2 Report
I think this is an important article, empirically grounded and theoretically nuanced. The importance of Fanon is well stated, as is the material on borderlands.
In the Abstract it might be useful to explain what you mean by 'racialized form of violence' - the body of your work implies that the violence was not on the side of the protesters, but concerns the response of authorities. At first glance a reader may be led to believe that you are claiming that the protest itself, in so far as it was directed as Rhodes, was itself a racialized form of violence, but this is indeed not at all what you are saying. The campaign itself must be defended against any claim that it has racist undertones. This, as you rightly make clear, is not at all the case.
Because the article is correctly anonymized it was not possible for me as reader to know the authorial voice. But normally you talk about 'we', and I think it would be useful at the beginning of the article to situate yourself(selves) in the narrative. Who are you, were you yourself participants in the protest, sympathetic observers or what?
Also I think 'Curating Violence' also needs some elucidation. As the article proceeds it becomes clear that you are talking about three events in which an artistic imput to the campaigns is being discuss (installations/events etc). In museum studies, 'curating' will be clear. But the casual reader may well be puzzled by what exactly 'curating violence' amounts to. Is the curating meant as part of the on-going campaign, or a commentary on the campaign. (or, of course, both). I thinkg that some elucidation of these issues at the begnning will help to put the valuable article in a clearer context.
Author Response
Thank you for the attentive and detailed review (05 April 2019). Herewith my response:
1. explain what you mean by 'racialized form of violence' - the body of your work implies that the violence was not on the side of the protesters, but concerns the response of authorities… The campaign itself must be defended against any claim that it has racist undertones. This, as you rightly make clear, is not at all the case.
Response: in the abstract I have adjusted the claim related to racialized forms of violence (see line 12). In the introduction, I have inserted two sentences that place these two campaigns in relation to longer (colonial and apartheid) histories of racialized exclusion from higher education in South Africa, and how resistance thereto is often met with racialized violence, in physical, epistemic and symbolic forms. (see lines 49-55)
2. it was not possible for me as reader to know the authorial voice. But normally you talk about 'we', and I think it would be useful at the beginning of the article to situate yourself(selves) in the narrative. Who are you, were you yourself participants in the protest, sympathetic observers or what?
Response: I have inserted an extended account of the ways that academic staff more generally got involved in support of the student campaigns (lines 113-144). In this passage of text, I also clarify my own position as author, and relation to my collaborators.
3. 'Curating Violence' also needs some elucidation. As the article proceeds it becomes clear that you are talking about three events in which an artistic input to the campaigns is being discuss (installations/events etc.). …Is the curating meant as part of the on-going campaign, or a commentary on the campaign. (or, of course, both). I think that some elucidation of these issues at the beginning will help to put the valuable article in a clearer context.
Response: In the section of the article that introduces the three curatorial events, I wrote a passage of text that clarifies my conception of curating as commentary, and as continuous pedagogic process related to the student campaigns. (see lines 289-296)